# The Essential Oil Derived from *Perilla frutescens* (L.) Britt. Attenuates Imiquimod–Induced Psoriasis-like Skin Lesions in BALB/c Mice

**DOI:** 10.3390/molecules27092996

**Published:** 2022-05-07

**Authors:** Yani Xu, Yaohui Shi, Jingxia Huang, Hongtao Gu, Chunlian Li, Lanyue Zhang, Guanting Liu, Wei Zhou, Zhiyun Du

**Affiliations:** The School of Biomedical and Pharmaceutical Sciences, Guangdong University of Technology, Guangzhou 510006, China; anneinee_she@163.com (Y.X.); 18344513188@163.com (Y.S.); hjx1007xia@163.com (J.H.); 2112112030@mail2.gdut.edu.cn (H.G.); 2112112016@mail2.gdut.edu.cn (C.L.); zhanglanyue@gdut.edu.cn (L.Z.); 3218001788@mail2.gdut.edu.cn (G.L.); guhongtao20140111@163.com (W.Z.)

**Keywords:** essential oil, psoriasis, inflammation, imiquimod

## Abstract

Psoriasis is reported to be a common chronic immune-mediated skin disease characterized by abnormal keratinocytes and cell proliferation. Perilla leaves are rich in essential oils, fatty acids, and flavonoids, which are recognized for their antioxidant and anti-inflammatory effects. In this study, the alleviating effect of essential oil (PO) extracted from *Perilla frutescens* stems and leaves on imiquimod (IMQ) -induced psoriasis-like lesions in BALB/c mice were investigated. Results showed that PO ameliorated psoriasis-like lesions in vivo, reduced the expression of lymphocyte antigen 6 complex locus G6D (Ly-6G), which is a marker of neutrophil activation, and inhibited the expression of inflammatory factors interleukin 1 (IL-1), interleukin 6 (IL-6), inducible nitric oxide synthase (iNOS), and cyclooxygenase 2 (COX2). In addition, PO significantly decreased the expression of cytokines such as IL-6, IL-1, interleukin 23 (IL-23), interleukin 17 (IL-17), and nuclear factor kappa-B (NF-κB). Furthermore, the down-regulation of mRNA levels of psoriasis-related pro-inflammatory cytokines, such as IL-17, interleukin 22 (IL-22), IL-23, interferon-α (IFN-α), and Interferon-γ (IFN-γ) was observed with the treatment of PO. All results show a concentration dependence of PO, with low concentrations showing the best results. These results suggest that PO effectively alleviated psoriasis-like skin lesions and down-regulated inflammatory responses, which indicates that PO could potentially be used for further studies on inflammation-related skin diseases such as psoriasis and for the treatment of psoriasis such as psoriasis natural plant essential oil resources.

## 1. Introduction

Psoriasis is an immune-mediated genetic disease that occurs in the skin or joints. Psoriasis presents many challenges, including high morbidity, chronicity, disease, disability, and associated comorbidities [1]. The common occurrence of psoriasis on the skin primarily involves an inflammatory response involving IFN-α, IFN-γ, IL-1, IL-6, IL-17, IL-22, IL-23 from dendritic cells, macrophages, and helper T cell (Th cells) [2,3]. In addition, the development of psoriasis is accompanied by an NF-κB-mediated inflammatory response involving two enzymes, iNOS, and COX2, in immune cells. Infiltration of neutrophils also occurs in psoriasis [4]. The pathogenesis of psoriasis is quite complex understanding the role of immune function in psoriasis, and the interactions between the innate and adaptive immune systems can help in the management of this complex disease.

Many natural essential oils have specific compounds with antibacterial, antioxidant, anti-inflammatory and anti-itch properties [5], making them attractive alternative and complementary therapies for dry and inflammatory skin conditions associated with skin barrier disruption. Natural essential oils have been studied for psoriasis, such as Sandal-wood Album Oil [6] and Bergamot essential oil [7]. *Perilla frutescens* (L.) Britt. is an annual herbaceous plant belonging to the family Lamiaceae. It plays an important role in traditional Chinese medicine, mainly used to treat cold, cough, nausea, vomiting, abdominal pain, constipation, asthma, and food poisoning [8]. As a traditional Chinese herbal medicine resource in Asia, it is widely distributed in China, Bhutan, India, Indonesia, Japan, Korea, and other Asian regions [9,10]. The stems and leaves of perilla are also often used as Chinese herbal medicine in traditional Chinese medicine treatment. The essential oils of its stems and leaves have been reported in many articles to be antibacterial, anti-inflammatory, and anti-oxidant, and are used in the food industry and the pharmaceutical industry [11,12]. So far, there is no research reported on the treatment of PO in psoriasis.

In this study, we adopted the IMQ-induce psoriasis model in BALB/c mice [13], and tacrolimus was used as a positive control and PO as the therapeutic agent. We found that PO was effective in alleviating the psoriasis-like symptoms and inflammatory response induced by IMQ, which indicates that PO may be an important natural plant essential oil resource for the treatment of psoriasis.

## 2. Materials and Methods

### 2.1. Material

IL-6, IL-1, TNF-α, IL-23, IL-17, and NF-κB Elisa Kits were purchased from Jiangsu Meibiao Biotechnology (Nanjing, China). IMQ cream and tacrolimus ointment was purchased from Sichuan Med-shine Pharmaceutical Co., Ltd. (Nanjing, China).

### 2.2. Essential Oil Extraction and Gas Chromatography and Mass Spectrometry (GC-MS)

The dried *Perilla frutescens* (L.) Britt. stems and leaves were purchased from DOCLAB (Guangzhou, China) in January 2021, and were washed, dried, sliced, and mechanically crushed. The powder was soaked with distilled water for 24 h. The essential oil was extracted through steam distillation and dehydrated with anhydrous sodium sulfate [14,15]. GC-MS method was used to analyze the compositions of PO [16].

### 2.3. Animals and Treatment

Total of 48 normal BALB/c mice (female, weighing 20 to 23 g, 6 to 8 weeks; Guangdong Medical Laboratory Center, Guangzhou, China) were shaved with an electronic razor and depilated with hair removal cream on the back of the skin two days prior to the formal experiment. All animal experiments were performed in accordance with relevant laws and regulations and were approved by the Laboratory Animal Center of Wuyi University (SCXK/2016-0041). All mice were randomly divided into the following groups, 8 mice per group: normal control group (NC, 60 mg/d Vaseline), model control group (MC, 62.5 mg/d 5% IMQ cream, 60 mg/d Vaseline), positive control group (PC, 62.5 mg/d 5% IMQ cream, 60 mg/d Tacrolimus ointment), low-dose PO group (PO-L, 62.5 mg/d 5% IMQ cream, 60 mg/d 0.01% PO diluted in Vaseline), medium-dose PO group (PO-M, 62.5 mg/d 5% IMQ cream, 60 mg/d 0.5% PO diluted in Vaseline), high-dose PO group (PO-H, 62.5 mg/d 5% IMQ cream, 60 mg/d 1% PO diluted in Vaseline). The skin condition of the back of all mice was recorded by camera each day before the formal experiment. About 62.5 mg/d 5% IMQ cream was applied uniformly to the back of all mice except the NC group. Four hours later, each group was applied with the Tacrolimus ointment, PO in Vaseline, or Vaseline to the back. The experiment lasted for 7 days. The weight of all mice was recorded, dorsal skin on the back, eye blood, and spleen were taken from the back of all mice on the 7th day.

### 2.4. Evaluation of the Severity of Psoriasis-like Skin Lesions

The severity of psoriasis-like lesions in mice was calculated according to a modified clinical psoriasis area and severity index (PASI) [17]. Erythema, scaling, and skin thickness were independently scored blindly on a scale from 0 to 4. 0, none; 1, mild; 2, moderate; 3, marked; 4, very marked. The total score of these three areas (from 0 to 12) was used as a thematic measure of psoriasis severity.

### 2.5. Spleen Index Calculation

The body weights of all mice were recorded and the spleens of all mice were acquired on the last day. The spleen index was calculated following the formula
$$\frac{\mathrm{Weight}\;\mathrm{of}\;\mathrm{the}\;\mathrm{spleen}}{\mathrm{Body}\;\mathrm{weight}\;\mathrm{of}\;\mathrm{mouse}}\times 10$$

The unit of the weight of the spleen: mg.

The unit of the bodyweight of the mouse: g.

### 2.6. Histopathological and Immunohistochemical Examination

The skin samples were fixed with 4% paraformaldehyde and embedded in paraffin at 4 °C overnight. About 4 μm-thick paraffin sections were cut out from the paraffin block embedded with skin tissue and loaded on a silane-coated glass slide for H&E staining. For immunohistochemistry, 4 μm-thick skin sections were deparaffinized, hydrated, treated with sodium citrate buffer pH 6.0 for antigen, and incubated overnight at 4 °C with a 1:50 dilution of purified rabbit antibody. Subsequently, sections were overlaid on a goat anti-rabbit IgG detection system. The image was taken with an automatic digital slide scanning system (Z1/AxioScan.Z1, ZEISS, Jena, Germany). The optical density of immunohistochemistry was analyzed with Image-Pro Plus 6.0 (MEDIA CYBERNETICS, Rockville, MD, USA).

### 2.7. RNA Extract and Real-Time Polymerase Chain Reaction

Total RNA was extracted from mouse skin samples using the RNA Extraction Solution (G3013, Servicebio, Wuhan, China) and RNA concentrations were measured using a spectrophotometer (NanoDrop2000, Thermo, Guangzhou, China). The synthesis of cDNA was performed under the instruction of Servicebio^®^RT First Strand cDNA Synthesis Kit (G3330, Servicebio, China). The PCR reactions were performed under the following thermocycling conditions: 10 min at 95 °C followed by 40 cycles of 15 s at 95 °C and 1 min at 60 °C 40 cycles at 60 °C. Expression levels were normalized to the gene expression levels of GAPDH. Relative gene expression was calculated using the 2^−ΔΔCT^ method. The specific primer sequences used for PCR are listed are listed in Table 1.

### 2.8. Enzyme-Linked Immunosorbent Assay (Elisa)

The determinations of IL-6, IL-1, TNF-α, IL-23, IL-17, and NF-κB were accomplished by the Elisa method with eye blood serum. Whole eye blood specimens were centrifuged at 1000× *g* for 20 min at room temperature for 2 h or 4 h, or overnight, and the supernatant was collected and stored at −80 °C in the refrigerator for Elisa assay. Elisa assays were performed under the instructions of the Elisa kit.

### 2.9. Statistical Analysis

All data are expressed as the mean ± standard deviation (x ± s). Statistical analysis was performed with Graph-Pad Prism 8.0 software. One-way analysis of variance was used for data analysis, followed by Tukey’s multiple comparisons test to compare differences among the groups. *p* < 0.05 was considered the threshold for statistical significance.

## 3. Results

### 3.1. GC–MS Analysis

The PO was analyzed by GC-MS as shown in Table 2, and 26 components were identified, accounting for 98.8% of the total essential oil. The relative contents of the PO components were determined by comparing the peak areas of the detected compounds with the ratio of the total areas of all detected compounds, with 2-Hexanoylfuran accounting for 42.15%, 2-(2-Methyl-1-propenyl) bicyclo[2.2.1]heptane for 18.61%, and Isocaryophyllene accounted for 13.02%, Caryophyllene oxide 6.44%, α-Farnesene 3.89%, α-Caryophyllene 1.52%, Germacrene D 1.44%. There was no significant difference in the content of other major substances. The differences in the major components may be due to a variety of factors, including various raw materials, origin, extraction, and detection methods.

### 3.2. PO Relieves IMQ-Induced Psoriasis-like Symptoms

IMQ applied to the dorsal skin for 6 days significantly induced epidermal scaling, erythema, inflammatory infiltration, and thickening in the dorsal side of mice compared to the NC group (Figure 1A). Notably, PO significantly reversed these psoriasis-like pathological changes. Objective blind scoring of erythema (Figure 1D), scaling (Figure 1E), and epidermal thickening (Figure 1F) on the back of mice were performed according to PASI principles. As shown in Figure 1A, a significant improvement in the erythema of the dorsal skin induced by IMQ, a reduction in scaling, and epidermal thickening, smoother skin were found in the PO-treated group. Thickened epidermis was obviously attenuated compared to those of the PC group treated with tacrolimus ointment. Surprisingly, the PO-L group even outperformed the PC group. As shown in PASI score (Figure 1G), the overall treatment effect of PO was ranked as PO-L, PO-M, and PO-H. These results indicate that PO is effective in relieving IMQ-induced psoriasis-like symptoms, and the therapeutic effect of PO is concentration-dependent.

The spleen index reflects the degree of lymphocyte proliferation in mice, which is related to immune response. In this study, the spleen index of all mice was calculated. The result showed that topical administration of IMQ significantly increased the spleen index of mice (*p* < 0.0001, Figure 1C), while PO treatment decreased the spleen index (*p* < 0.01).

### 3.3. PO Inhibited Neutrophil Activation and the Expression of Inflammation-Related Factors

Ly-6G is a cell surface marker of neutrophils, and the proliferation of neutrophils reflects a decent immune process in mice. It has been reported that topical application of IMQ ointment can induce significant proliferation of neutrophils [24]. IMQ induces the activation of macrophages that synthesized iNOS and COX2 associated with the production of inflammatory factors such as IL-1, IL-6, and it has been found psoriatic lesions may lead to high levels of iNOS expression [25,26] and COX2 [27]. Therefore, the observation of IL-1, IL-6, iNOS, and COX2 is important to detect the immune response and inflammation process. In this study, IL-1, IL-6, INOS, COX2 were detected by immunohistochemistry (Figure 2A) and all key factors were quantitatively analyzed both in full skin as well as in epidermis (Figure 2B–E). As it is shown in Figure 2, the expression level of Ly-6G was significantly increased both in the full skin (*p* < 0.0001, Figure 2I) and in the epidermis (*p* < 0.0001, Figure 2H) after the irritation of IMQ in the MC group. With the application of PO, an inhibitory effect of PO on Ly-6G expression (*p* < 0.0001) was found with the best in the PO-L group. The expression of both iNOS (*p* < 0.0001, Figure 2F) and COX2 (*p* < 0.0001, Figure 2J) were significantly higher in the MC group compared to the NC group, especially in the epidermis (*p* < 0.0001, Figure 2G for iNOS, Figure 2K for COX2). The elevated expression of both iNOS and COX2 in the tissues suggested increased synthesis of inflammatory factors. From the results, it could be seen that the expression of IL-1 (*p* < 0.0001, Figure 2B) and IL-6 (*p* < 0.0001, Figure 2D) was also significantly higher in the MC group and the difference in expression in the epidermis was more pronounced compared to the NC group (*p* < 0.0001, Figure 2C,E), which is consistent with the immunohistochemical results of iNOS and COX2. With the treatment of PO, the expression of iNOS and COX2 in the tissues was significantly reduced, accompanied by suppression of IL-1 and IL-6 expression. Among all the treatment groups, the PO-L group showed the best inhibition of iNOS, COX2, IL-1, and IL-6 in both the full skin and epidermis. These results suggest that PO-L and PO-M has a surprising inhibitory effect on iNOS, COX2, IL-1, and IL-6 associated with the inflammatory response induced by IMQ in the skin.

### 3.4. PO Downregulates mRNA Levels of Immune-Related Factors in Skin Tissues

Studies have shown that IMQ induces the activation of dendritic cells to produce IFN-α [28] and IFN-γ [29], which stimulates the activation of macrophages to produce the inflammatory factor IL-23 and the activation of T-lymphocyte (T) cells to produce IL-22 and IL-17 [30]. Elevated expression levels of IFN-α, IFN-γ, IL-22, and IL-17 have an exacerbating effect on the skin epidermis by stimulating an increased inflammatory response, which can exacerbate psoriasis-like symptoms in the skin. In this study, real time polymerase chain reaction (RT-PCR) was used to determine the mRNA levels of IFN-α, IFN-γ, IL-22, IL-23, and IL-17. IFN-α, IFN-γ, IL-22, IL-23, and IL-17 in skin tissue were significantly increased with the topical application of IMQ (*p* < 0.0001 or *p* < 0.05) as shown in Figure 3. Furthermore, the treatment of the PO-L group had a comparable effect to that of receiving the positive drug tacrolimus in the PC group. These results suggest that PO can inhibit the expression of factors associated with psoriasis-related immune responses at the mRNA level, which suggests that PO is effective in the remission of psoriasis.

### 3.5. PO Inhibits the Expression of Inflammatory Factors in IMQ-Induced Mice in Serum

The development of psoriasis is mediated by dendritic cells as well as lymphocytes and leukocytes. Dendritic cells and macrophages produce inflammatory cytokines such as interleukin-23 (IL-23), interleukin-1 (IL-1), interleukin-6 (IL-6) [31], which promote the differentiation of TH17 cells to secrete IL-17 and other mediators that stimulate epidermal cell proliferation and promote the activation of NF-κB [32]. NF-κB is involved in the development of psoriatic lesion with elevated levels in the blood and is involved in the inflammatory process [33]. In the present study, Elisa was used to determine IL-1, IL-6, IL-17, IL-22, and IL-23 in the serum of all mice. From the experimental results, it can be seen that the levels of IL-1, IL-6, IL-23, IL-17, and NF-κB were significantly higher in the MC group in serum (*p* < 0.05, *p* < 0.01, *p* < 0.0001, Figure 4). After topical application of PO, the expression level of IL-1, IL-6, IL-23, IL-17, and NF-κB in serum was respectively reduced. These results indicate that PO has a surprising inhibitory effect on IL-1, IL-6, IL-23, IL-17, and NF-κB.

## 4. Discussion

Psoriasis is a long course immune system-related disease involving complex mechanisms associated with excessive production of inflammatory factors, excessive proliferation, and keratinization of keratinocytes in the epidermal layer of the skin [34]. The main first-line therapeutic agents currently used in the clinical management of psoriasis include methotrexate [35], cyclosporine [36], and avidin (etretinate), which may cause allergic reactions in humans. Therefore, it is important to find a natural plant resource with therapeutic effects for psoriasis. In studies, the essential oil of the leaves of *Perilla frutescens*, a natural medicinal resource, has been validated for its antioxidant, antibacterial, and anti-inflammatory activities [37,38,39]. Therefore, we hypothesize that PO can treat psoriasis by modulating keratinocyte hyperproliferation keratinization and inflammatory responses. In this study, we investigated the effect and mechanism of PO on the alleviation of psoriasis-like symptoms induced by IMQ, and it was the first time that PO was applied to the anti-psoriasis model, and there is no relevant research. Further research on this basis is of great significance to expand the medical and commercial application of Perilla essential oil.

In this study, the top five components with the highest content in PO (Table 2) are respectively 2-hexanoylfuran, 2-(2-METHYL-1-propenyl) Bicyclo[2.2.1]heptane, Isocar-yophyllene, Caryophyllene oxide, and Linalool, which are roughly the same as those of published PO in GC-MS analysis. The discrepancy in the content of specific components of Linalool or 2-acetylfuran, which are the main components [20], may be caused by various factors such as plant origin, testing equipment, etc. It has been reported that 2-hexanoylfuran has anti-allergic and anti-dermatitis properties [40], and Linalool has anti-oxidant and anti-inflammatory activities [41]. The availability of PO for psoriasis re-mission may be related to the action of these two components.

IMQ-induced psoriasis is the most commonly used disease model to study psoriasis. In the IMQ-induced psoriasis model, elevated expression levels of cytokines such as IL-17, IL-6, and IL-1 are attributed to the proliferation of macrophages, dendritic cells, mast cells, and neutrophils, and the increase in these factors suggests an inflammatory response [42,43]. IMQ is an agonist of Toll-like receptor (TLR) 7/8, which induces an immune response in the body [44]. The rapid proliferation of immune cells may lead to splenomegaly. In addition, IMQ induces hyperproliferative keratinization in the epidermis and promotes the release of inflammatory factors such as IL-1 and IL-6, leading to psoriasis-like symptoms in the skin, which includes erythema, scaling, and epidermal thickening [45]. PO was scored according to the PASI principles for the relief of psoriasis-like symptoms. The results showed that PO was effective in reducing erythema and scaling of the skin and in inhibiting the thickening of the cuticle, with 0.01% of PO daily treatments being even more effective than the tacrolimus treatment group (Figure 1). The spleen size of the mice was also restored from enlarged to near normal after PO treatment (Figure 1C). In the IMQ-induced immune response, dendritic cells were activated by IMQ binding to Toll-like receptors and secreted IFN-α, which stimulated T-cell activation [46]. T cells further produced IL-17 and IL-22 for transport to the epidermis of the skin, which promoted the secretion of inflammatory factors IL-1 and IL-6 from the epidermis as well as exacerbating skin keratinization [47]. Furthermore, IFN-γ secreted by Th cells promotes the proliferation of leukocytes to activate the inflammatory response, increases the enzymatic synthesis of iNOS and COX2, and promotes the secretion of IL-1 and IL-6 by leukocytes [48,49,50]. IL-23 produced by leukocytes enhances T-cell activation and immune response, and IL-22 and IL-23 produced by T-cell activation stimulate neutrophil proliferation and cause neutrophil infiltration in the skin. We examined inflammatory factors and the neutrophil cell surface marker Ly-6G in skin tissue using immunohistochemistry. PO-L effectively inhibited the expression of iNOS, COX2, IL-1, and IL-6 in the skin (Figure 2), suggesting that PO could alleviate the inflammatory response in skin tissue. PO showed positive anti-inflammatory activity in the present study which is consistent with some previous studies in which it was able to inhibit the LPS-mediated inflammatory and NO-producing responses of RAW 264.7 cells [51,52,53]. In addition, the expression of Ly-6G was significantly decreased in the skin, which means that the number of neutrophils decreased and PO was able to inhibit the proliferation of neutrophils. From the experimental results in Figure 3, it can be seen that PO inhibited the mRNA levels of IFN-α, IFN-γ, IL-22, IL-23, and IL-17, which are key factors in psoriasis, indicating that PO was able to block the proliferation of neutrophils.

IL-6, IL-1, IL-23, IL-17, and NF-κB in serum are secreted into the serum by Th cells. NF-κB is a key transcription factor responsible for the pathogenesis of psoriasis that regulates immunity, proliferation, and apoptosis, as well as triggering the production of pro-inflammatory cytokines such as TNF-α and IL-17. It can also trigger the production of pro-inflammatory cytokines, such as IL-17. Elisa results (Figure 4) showed that PO was effective in reducing serum levels of NF-κB, as well as IL-1, IL-6, IL-17, and IL-23.

## 5. Conclusions

In conclusion, our study reveals that PO improves IMQ-induced psoriasis. The effects of PO treatment are manifested by histological changes, PASI scores, and splenic indices. PO downregulates the immune cytokines IL-17, IL-22, IL-23, IFN-α, IFN-γ, and pro-inflammatory cytokines such as IL-IL-6 and IL-1 in a dose-dependent manner, which showed the best effect in low-dose. PO is a potential source of topical natural essential oils for the treatment of psoriasis, as it alleviates both the immune and inflammatory responses to psoriasis.

## Figures and Tables

**Figure 1 molecules-27-02996-f001:**
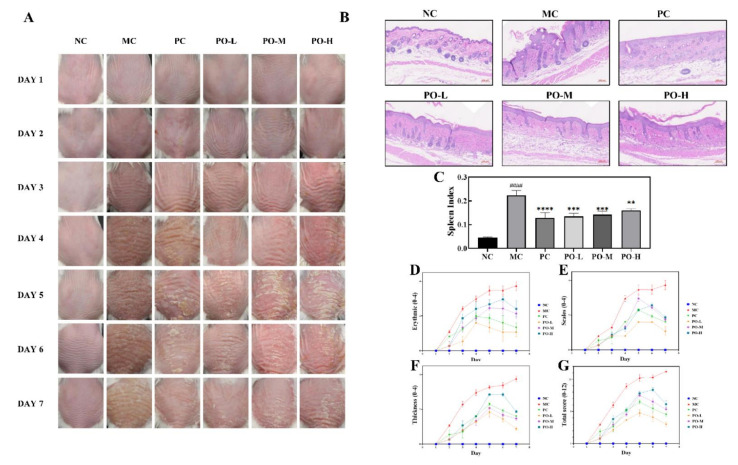
PO ameliorated imiquimod-induced psoriasis in BALB/c mice (PO, 0.01%, 0.5%, 1%, 60 mg/kg/d). (**A**) Skin changes record of BALB/c mice every day during the experiment. (**B**) H&E staining of BALB/c mice. (**C**) The spleen index of each group. The spleen index was calculated following the formula Spleen index = (spleen weight/mouse weight) × 10. (**D**–**G**) Erythema, scales, skin thickness, and total scores were evaluated daily based on the modified Psoriasis Area and Severity Index (PASI). Data are shown as mean ± SEM (n = 8), #### *p* < 0.0001 vs. NC group, ** *p* < 0.01, *** *p* < 0.001, **** *p* < 0.0001 vs. MC group.

**Figure 2 molecules-27-02996-f002:**
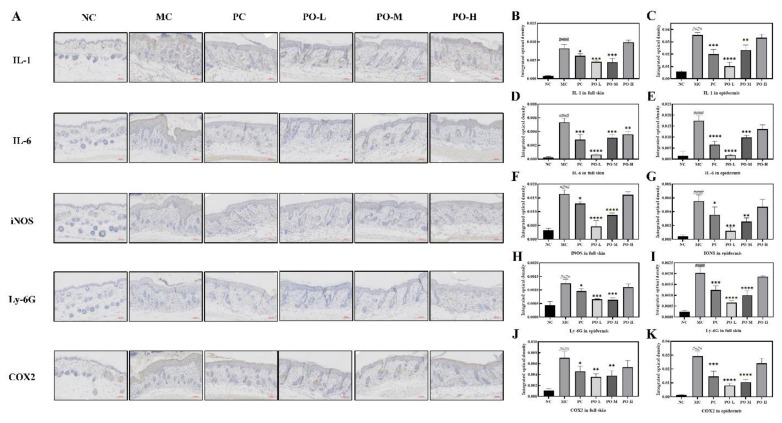
PO inhibited the expression of Inflammation-related factors (IL-1, IL-6, iNOS, COX2) and neutrophil maturation marker (Ly-6G). (**A**) Immunohistochemistry of IL-1, IL-6, iNOS, Ly-6G, and COX2. The target protein is stained tan. (**B**,**D**,**F**,**H**,**J**) The quantitative analysis of IL-1, IL-6, iNOS, Ly-6G, and COX2 in full skin. (**C**,**E**,**G**,**I**,**K**) The quantitative analysis of IL-1, IL-6, iNOS, Ly-6G, and COX2 in the epidermis. All quantitative analysis were performed with Image-Pro Plus 6.0 (MEDIA CYBERNETICS, USA). Data are shown as mean ± SEM (n = 8), #### *p* < 0.0001 vs. NC group, * *p* < 0.05, ** *p* < 0.01, *** *p* < 0.001, **** *p* < 0.0001 vs. MC group.

**Figure 3 molecules-27-02996-f003:**
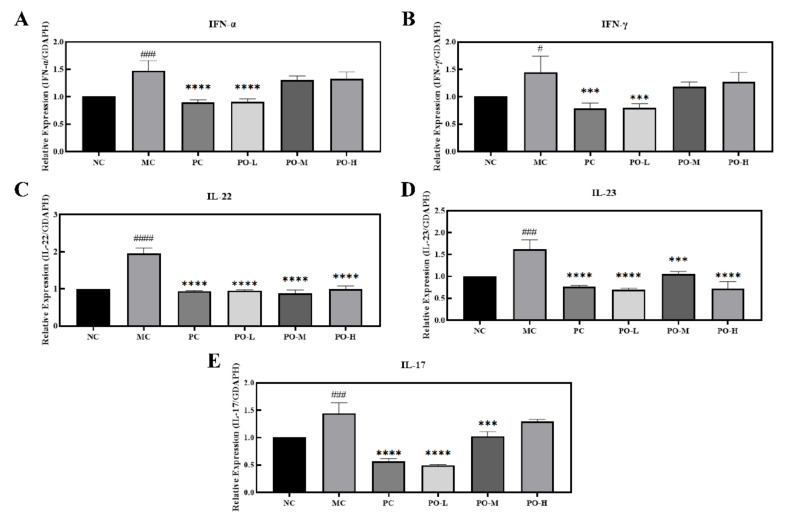
PO down-regulated the mRNA levels of IFN-α, IFN-γ, IL-22, IL-23, and IL-17 in the skin of BALB/c mice. (**A**) mRNA level of IFN-α. (**B**) mRNA level of IFN-γ. (**C**) mRNA level of IL-22. (**D**) mRNA level of IL-23. (**E**) mRNA level of IL-17. The internal reference for all target mRNAs is GADPH. Data are shown as mean ± SEM (n = 8), # *p* < 0.05, ### *p* < 0.001, #### *p* < 0.0001 vs. NC group, *** *p* < 0.001, **** *p* < 0.0001 vs. MC group.

**Figure 4 molecules-27-02996-f004:**
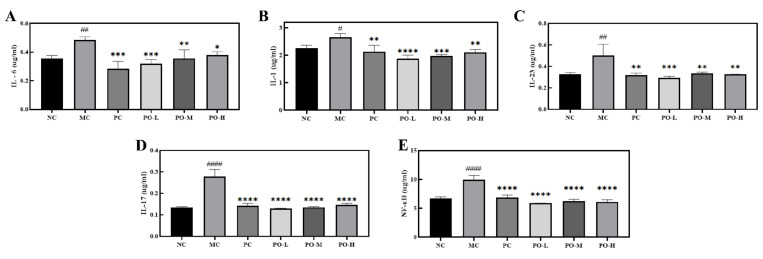
The inhibitory effect of PO on the expression level of inflammation-related factors and the nuclear factor kappa-B. **(A)** The expression level of IL-6 in the serum. (**B**) The expression level of IL-1 in serum. (**C**) The expression level of IL-23 in the serum. (**D**) The expression level of IL-17 in the serum. (**E**) The expression level of NF-κB in serum. IL-6, IL-1, IL-23, IL-17, and NF-κB in serum of BABL/c mice were determined with Elisa kits. Data are shown as mean ± SEM (n = 8), # *p* < 0.05, ## *p* < 0.01, #### *p* < 0.0001 vs. NC group, * *p* < 0.05, ** *p* < 0.01, *** *p* < 0.001, **** *p* < 0.0001 vs. MC group.

**Table 1 molecules-27-02996-t001:** The sequences of primers.

Gene	Forward Primers	Reverse Primers
IL-17	ATCTGTGTCTCTGATGCTGTTG	CGTGGAACGGTTGAGGTAGTCT
IL-22	CTCACCGTGACGTTTTAGGGA	CCACCATAGGAGGCCACAAG
IL-23	AGACTAAAAATAATGTGCCCCG	GCTATCAGGGAGTAGAGCAGGC
IFN-α	CCTGCTGGCTGTGAGGAAATAC	ACTTCTGCTCTGACCACCTCCC
IFN-γ	CCATCGGCTGACCTAGAGAAGAC	GCCACTTGAGTTAAAATAGTTATTCAGAC
GAPDH	CCTCGTCCCGTAGACAAAATG	TGAGGTCAATGAAGGGGTCGT

**Table 2 molecules-27-02996-t002:** Retention Index (RI) and relative content (%) of chemical compositions identified from *Perilla frutescens* essential oils.

No	Compounds ^i^	RI ^ii^	Exp. RI	Ref.	Relative Content
					*Perilla frutescens*
1	3-Octenol	1004	972		0.11
2	3-Octanol	1011	1393	a	0.14
3	Linalool	1086	1104	b	5.45
4	2,2-Dimethyl-3-heptanone	1176	965		0.53
5	borneol	1183	1165	b	0.1
6	2-Cyclopenten-1-one, 2-(2-butenyl)-3-methyl-, (*Z*)-	1186	1183	c	1.05
7	2-Hexanoylfuran	1207	1283	d	42.15
8	3,4-Dihydro-5-methyl-2H-pyran-4-carboxylic acid ethyl ester	1215			0.12
9	2-(2-Methyl-1-propenyl)bicyclo[2.2.1]heptane	1241			18.61
10	2,6-Octadienoic acid, 3,7-dimethyl-, methyl ester	1248	1326		0.16
11	Copaene	1276	1387	e	0.59
12	β-Bourbonene	1283	1377	b	0.22
13	β-Elemene	1289	1398	c	0.48
14	Isocaryophyllene	1303	1407	c	13.02
15	α–Caryophyllene	1329	1452	f	1.52
16	Germacrene D	1347	1480	f	1.44
17	α–Farnesene	1352	1508	d	3.89
18	γ-Muurolene	1368	1456	c	0.1
19	δ-Cadinene	1375	1524	c	0.56
20	Nerolidol	1413	1565	b	0.87
21	Caryophyllene oxide	1445	1578	b	6.44
22	Cedrol	1447	1589		0.13
23	Humulene epoxide II	1452	1605	c	0.46
24	α-Cadinol	1490	1653	c	0.34
25	Viridiflorol	1508	1590	c	0.18
26	Hexahydrofarnesyl acetone	1619	1846	f	0.14
	Total identified				98.8
	Total monoterpenoids/%				42.15
	Oxygenated monoterpenes/%				42.15
	Total sesquiterpenoids/%				26.98
	Sesquiterpene hydrocarbons/%				20.08
	Oxygenated sesquiterpenes/%				6.9
	Others/%				2.69

^i^ Compound listed in the order of elution from methyl silicone capillary column (30 m × 0.25 mm, 0.25-μm film thickness). ^ii^ Retention indices (RIs) relative to n-alkanes (C_6_–C_40_) on the same methyl silicone capillary column. a Yue et al. (2021) [18]; b Chen et al. (2019) [19]; c Tian et al. (2014) [20]; d Zhang et al. (2018) [21]; e Huang et al. (2011) [22]; f Liu et al. (2012) [23].

## Data Availability

The data presented in this study are available on request from the corresponding author. The data are not publicly available due to funding project requirements.

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
