# Peer review of "The Essential Oil Derived from Perilla frutescens (L.) Britt. Attenuates Imiquimod–Induced Psoriasis-like Skin Lesions in BALB/c Mice"

_molecules, 2022, doi:10.3390/molecules27092996_

Round 1
Reviewer 1 Report
This is an interesting research activity in which combined experimental and computational methods have been used for the evaluation of The essential oil derived from Perilla frutescens (L.) Britt. atten- 2
uates imiquimod–induced psoriasis-like skin lesions in BALB/c 3
mice
The following corrections can enrich the paper:
1. Introduction Should be revised and updated by new references.
2. How does this work differ from the previously reported several classes of this topic??? Novelty of the study should be further highlighted.
3. It is recommended to compare the results of this study with those reported in literature especially in this journal.
4. Did you compare the results of this study with your recently published ones?
5. Kindly update following recent references in the article:
Antibacterial activity and virtual screening by molecular docking of lycorine from Pancratium foetidum Pom (Moroccan endemic Amaryllidaceae)
Microbial Pathogenesis, Volume 115, February 2018, Pages 138-145
H. Bendaif, A. Melhaoui, M. Ramdani, Y. El Ouadi
Antibacterial, antifungal and antioxidant activity of total polyphenols of Withania frutescens.L
Bioorganic Chemistry, Volume 93, December 2019, Article 103337
Abdelfattah El Moussaoui, Fatima Zahra Jawhari, Ahmed M. Almehdi, Amina Bari
Author Response
Dear Editors and Reviewers,
We appreciate and acknowledge the comments and suggestions given by the reviewers in our manuscript entitled “The essential oil derived from Perilla frutescens (L.) Britt. attenuates imiquimod–induced psoriasis-like skin lesions in BALB/c mice” (Manuscript ID: molecules-1662868). Those comments are valuable and very helpful for the improvement of our paper as well as significant in guiding our research. Reviewers’ comments are written in black and our replies are written in blue. We have made corrections and highlighted the changes in red in the revised version of the manuscript and figure.
We hope, with the modifications and improvements based on your suggestions and other reviewers’ comments, the quality of our manuscript would meet the publication standard of Molecule. The major corrections in our revised manuscript and our point-by-point responses to the comments are as follows. If you have any question, please contact us without hesitate. We are looking forward to hearing from you.
Responses to the reviewers’ comments are as following.
Comments to the authors from Reviewer 1
- Introduction Should be revised and updated by new references.
Response: The introduction has revised and updated by new references.
- How does this work differ from the previously reported several classes of this topic??? Novelty of the study should be further highlighted.
Response: The novelty of the study has been highlighted in the introduction as “The essential oils of its stems and leaves have been reported in many articles to be anti-bacterial, anti-inflammatory, and anti-oxidant, and are used in the food industry and the pharmaceutical industry. So far, there is no research reported on the treatment of PO in psoriasis.” (Lines 52-55) and has been further discussed the novelty of the manuscript in the discussion: and it was the first time that PO was applied to the anti-psoriasis model, and there is no relevant research. Further research on this basis is of great significance to expand the medical and commercial application of Perilla essential oil. (Lines 270-273)
- It is recommended to compare the results of this study with those reported in literature especially in this journal.
Response: We have compared the results of this study with those reported in literature in this Molecule in the manuscript: PO showed positive anti-inflammatory activity in the present study is consistent with some previous studies in which it was able to inhibit the LPS-mediated inflammatory and NO-producing responses of RAW 264.7 cells. (Lines 309-312)
- Did you compare the results of this study with your recently published ones?
Response: We have compared the results of this study with those reported in literature in this Molecule in the manuscript: PO showed positive anti-inflammatory activity in the present study is consistent with some previous studies in which it was able to inhibit the LPS-mediated inflammatory and NO-producing responses of RAW 264.7 cells. (Lines 309-312)
- Kindly update following recent references in the article:
Antibacterial activity and virtual screening by molecular docking of lycorine from Pancratium foetidum Pom (Moroccan endemic Amaryllidaceae)
Microbial Pathogenesis, Volume 115, February 2018, Pages 138-145
- Bendaif, A. Melhaoui, M. Ramdani, Y. El Ouadi
Antibacterial, antifungal and antioxidant activity of total polyphenols of Withania frutescens.L
Bioorganic Chemistry, Volume 93, December 2019, Article 103337
Abdelfattah El Moussaoui, Fatima Zahra Jawhari, Ahmed M. Almehdi, Amina Bari
Response:Updated. (References 49, 50)
Reviewer 2 Report
In this study, the essential oil obtained from stems and leaves of Perilla frutescens (PO) was applied on imiquimod (IMQ) -induced psoriasis-like lesions in BALB/c mice. The oil effect was investigated on several levels: evaluation of the severity of psoriasis-like skin lesions, spleen index calculation, histopathological and immunohistochemical examination, real-time polymerase chain reaction and enzyme-linked immunosorbent assay.
The concept of this paper, the conducted experiments and the obtained results are excellent. Considering that the effect of PO oil on psoriasis lesions has not been studied so far, the results are completely new and promising for further treatment of this disease. I suggest that the manuscript be accepted with minor modifications
Lines 15-16 Results showed that PO ameliorated- Based on the obtained results, the effect of the oil was dose-dependent, and the lowest concentration of oil was the most effective. This should be emphasized in the abstract
Lines 27-31 Too many keywords, write only a few of the most important ones
Lines 138-139 The differences in the major components may be due to a variety of factors, including various raw materials, origin, extraction, and detection methods- This sentence should be in the discussion. There is no part in the discussion that refers to the chemical analysis of PO oil. It would be good if the authors would compare the chemical composition of the oil used in the work with the already published chemical composition of PO oil. Are there any data on the anti-inflammatory properties of the main chemical compounds detected in the used oil? If there are, this should be stated in the discussion.
Lines 200-202 These results suggest that PO has a surprising inhibitory effect on iNOS, COX2, IL-1, and IL-6 associated with the inflammatory response induced by IMQ in the skin- Instead of PO, write that the results refer to the groups PO-L and PO-M
Lines 283-284 PO effectively inhibited the expression of iNOS, COX2, IL-1, and IL-6 in the skin (Fig. 2), suggesting that PO could alleviate the inflammatory response in skin tissue-PO in lower concentrations
Technical errors
Line 65 the essential oil was ex- Capital letter at the beginning of the sentence
Line 136 a-Caryo- alpha, In Table 2 also change this mistake (a–Caryophyllene, a–Farnesene)
Author Response
Dear Editors and Reviewers,
We appreciate and acknowledge the comments and suggestions given by the reviewers in our manuscript entitled “The essential oil derived from Perilla frutescens (L.) Britt. attenuates imiquimod–induced psoriasis-like skin lesions in BALB/c mice” (Manuscript ID: molecules-1662868). Those comments are valuable and very helpful for the improvement of our paper as well as significant in guiding our research. Reviewers’ comments are written in black and our replies are written in blue. We have made corrections and highlighted the changes in red in the revised version of the manuscript and figure.
We hope, with the modifications and improvements based on your suggestions and other reviewers’ comments, the quality of our manuscript would meet the publication standard of Molecule. The major corrections in our revised manuscript and our point-by-point responses to the comments are as follows. If you have any question, please contact us without hesitate. We are looking forward to hearing from you.
Responses to the reviewers’ comments are as following.
Comments to the authors from Reviewer 2
- Lines 15-16 Results showed that PO ameliorated- Based on the obtained results, the effect of the oil was dose-dependent, and the lowest concentration of oil was the most effective. This should be emphasized in the abstract
Response: We have emphasized the concentration dependence of PO in abstract.: All results show a concentration dependence of PO, with low concentrations showing the best results. (Lines 23-24)
- Lines 27-31 Too many keywords, write only a few of the most important ones
Response: We have limited the keywords to four: Essential oil; Psoriasis; Inflammation; Imiquimod; (Line 29)
- Lines 138-139 The differences in the major components may be due to a variety of factors, including various raw materials, origin, extraction, and detection methods- This sentence should be in the discussion. There is no part in the discussion that refers to the chemical analysis of PO oil. It would be good if the authors would compare the chemical composition of the oil used in the work with the already published chemical composition of PO oil. Are there any data on the anti-inflammatory properties of the main chemical compounds detected in the used oil? If there are, this should be stated in the discussion.
Response: We have compared the chemical composition of the oil used in the work with the already published chemical composition of PO oil, and the biological activities of the main components were stated in the manuscript: In this study, the top five components with the highest content in PO (Table. 2) are respectively 2-hexanoylfuran, 2-(2-METHYL-1-propenyl) Bicyclo [2.2.1] Heptane, Isocar-yophyllene, Caryophyllene oxide and Linalool, which are roughly the same as those of published PO in GC-MS analysis. The discrepancy in the content of specific components of Linalool or 2-acetylfuran, which are the main components, may be caused by vari-ous factors such as plant origin, testing equipment, etc. It's been reported that 2-hexanoylfuran have anti-allergic and anti-dermatitis properties, and Linalool has anti-oxidant and anti-inflammatory activities. The availability of PO for psoriasis re-mission may be related to the action of these two components. (Lines 274-282)
- Lines 200-202 These results suggest that PO has a surprising inhibitory effect on iNOS, COX2, IL-1, and IL-6 associated with the inflammatory response induced by IMQ in the skin- Instead of PO, write that the results refer to the groups PO-L and PO-M
Response: Revised. (Line 206)
- Lines 283-284 PO effectively inhibited the expression of iNOS, COX2, IL-1, and IL-6 in the skin (Fig. 2), suggesting that PO could alleviate the inflammatory response in skin tissue-PO in lower concentrations
Response: Revised. (Line 307)
- Line 65 the essential oil was ex- Capital letter at the beginning of the sentence
Response: Revised. (Line 69)
- Line 136 a-Caryo- alpha, In Table 2 also change this mistake (a–Caryophyllene, a–Farnesene)
Response: Revised as “α–Caryophyllene, α–Farnesene”. (Line 143, Table 2)
Reviewer 3 Report
After a complete revision, the manuscript is ACCEPTED AFTER MINOR REVISION. In general, the study is connected to the journal's objectives. The study is very interesting and can contribute to the validation of some uses of essentials oils in traditional medicine and can give preliminary approaches to the use of essential oils for the treatment of diseases to autoimmune disease as psoriasis. However, the manuscript has some parts that need to be corrected.
In the following paragraphs, I give a detailed revision.
Best Regards
1. Introduction: It is essential to include a little more information about the traditional uses of Perilla; in what kind of diseases have been used?
2. Results: The graph's legend inside Figure 1 is unclear; it is necessary to increase the resolution.
3. Results: I suggest including the graphs in figure 2 in vertical distribution; in this way, the authors can increase their size. The way they are presented is not easy to visualize.
It is fundamental to include the Institutional Review Board Statement and approval number of the study since they worked with animals.
Author Response
Dear Editors and Reviewers,
We appreciate and acknowledge the comments and suggestions given by the reviewers in our manuscript entitled “The essential oil derived from Perilla frutescens (L.) Britt. attenuates imiquimod–induced psoriasis-like skin lesions in BALB/c mice” (Manuscript ID: molecules-1662868). Those comments are valuable and very helpful for the improvement of our paper as well as significant in guiding our research. Reviewers’ comments are written in black and our replies are written in blue. We have made corrections and highlighted the changes in red in the revised version of the manuscript and figure.
We hope, with the modifications and improvements based on your suggestions and other reviewers’ comments, the quality of our manuscript would meet the publication standard of Molecule. The major corrections in our revised manuscript and our point-by-point responses to the comments are as follows. If you have any question, please contact us without hesitate. We are looking forward to hearing from you.
Responses to the reviewers’ comments are as following.
Comments to the authors from Reviewer 3
- Introduction: It is essential to include a little more information about the traditional uses of Perilla; in what kind of diseases have been used?
Response: The traditional uses of Perilla and the diseases in which it has been used has been added to the Introduction: and it plays an important role in traditional Chinese medicine, mainly used to treat cold, cough, nausea, vomiting, abdominal pain, constipation, asthma and food poisoning as. (Lines 47-49)
- Results: The graph's legend inside Figure 1 is unclear; it is necessary to increase the resolution.
Response: The resolution of the Figure 1 has been improved.
- Results: I suggest including the graphs in figure 2 in vertical distribution; in this way, the authors can increase their size. The way they are presented is not easy to visualize.
Response: The arrangement and presentation of the figure 2 have been rearranged.
- It is fundamental to include the Institutional Review Board Statement and approval number of the study since they worked with animals.
Response: The Institutional Review Board Statement and approval number have been added to the manuscript: All animal experiments were performed in accordance with relevant laws and regulations and were approved by the Laboratory Animal Center of Wuyi University (SCXK/2016-0041). (Lines 76-78)